# Optimization Method for Phenolic Compounds Extraction from Medicinal Plant (*Juniperus procera)* and Phytochemicals Screening

**DOI:** 10.3390/molecules26247454

**Published:** 2021-12-09

**Authors:** Abdalrhaman M. Salih, Fahad Al-Qurainy, Mohammad Nadeem, Mohamed Tarroum, Salim Khan, Hassan O. Shaikhaldein, Abdulrahman Al-Hashimi, Alanoud Alfagham, Jawaher Alkahtani

**Affiliations:** Botany and Microbiology Department, College of Science King Saud University, P.O. Box 2455, Riyadh 11451, Saudi Arabia; falqurainy@ksu.edu.sa (F.A.-Q.); mnadeem@ksu.edu.sa (M.N.); mtarroum@ksu.edu.sa (M.T.); skhan2@ksu.edu.sa (S.K.); hassanbb2@gmail.com (H.O.S.); aalhashimi@ksu.edu.sa (A.A.-H.); aalfaghom@ksu.edu.sa (A.A.); Jsalqahtani@ksu.edu.sa (J.A.)

**Keywords:** flavonoid, tannin, solvents, spectrophotometry, chromatography analysis, phytochemicals profiling, new approach

## Abstract

*Juniperus procera* is a natural source of bioactive compounds with the potential of antitumor, antimicrobial, insecticidal, antifungal, and antioxidant activities. An optimization method was developed for total phenolic content (TPC), total flavonoid content (TFC), and total tannin content (TTC) in leaf and seed extract of *Juniperus procera.* Organic solvents (methanol (99.8%), ethanol (99%), and acetone (99.5%)), and deionized water (DI) were used for extraction. The estimation of TPC, TFC, and TTC in plant materials was carried out using UV-spectrophotometer and HPLC with the standards gallic acid, quercetin, and tannic acid. Recovery of TPC in leaf extract ranged from 2.9 to 9.7 mg GAE/g DW, TFC from 0.9 to 5.9 mg QE/g DW, and TTC ranged from 1.5 to 4.3 mg TA/g DW while the TPC value in the seed extract ranged from 0.53 to 2.6 mg GAE/g DW, TFC from 0.5 to 1.6 mg QE/g DW, and TTC ranged from 0.5 to 1.4 mg TA/g DW. This result revealed that methanol is the best solvent for recovery of the TPC value (9.7 mg) from leaf extract in comparison to other solvents. Ethanol recorded the highest result of TFC (5.9 mg) in leaf extract among the solvents whereas acetone was the best for TTC yield recovery from leaf extract (4.3 mg). In the case of the seed extract, ethanol was the best solvent for both TPC (2.6 mg), and TFC (1.6 mg) recovery in comparison to other solvents. Total tannin content in methanol resulted in significant recovery from seed extract (1.4 mg). Separation and quantification of gallic acid, quercetin, and tannic acid in plant materials were undertaken using HPLC. Gallic acid in leaf and seed of *J. procera* ranged from 6.6 to 9.2, 6.5 to 7.2 µg/g DW, quercetin from 6.3 to 18.2, 0.9 to 4.2 µg/g DW, and tannic acid from 16.2 to 29.3, 6.6 to 9.3 µg/g DW, respectively. Solvents have shown a significant effect in the extraction of phenolic compounds. Moreover, phytochemicals in plant materials were identified using GC-MS and resulted in very important bioactive compounds, which include anti-inflammatory, antibacterial, and antitumor agents such as ferruginol, phenanthrene, and n-hexadecanoic acid. In conclusion, the optimal solvent for extraction depends on the part of the plant material and the compounds that are to be isolated.

## 1. Introduction

*Juniperus procera* is a source of natural bioactive compounds with the potential of anti-tumor, antimicrobial, insecticidal, antifungal, and antioxidant activities [1,2,3,4]. In context, methanolic stem extracts of *J. procera* have been shown to exhibit antifungal effects against *Aspergillus flavus* growth and its mycotoxins [5]. Phenolic constituents are parts of secondary metabolites that are mostly found in different species of plants with huge structural diversities and can be exist as glycosides or aglycones [6,7]. Additionally, it has been reported that phenolic compounds are one of the major and diverse groups of phytochemical constituents in plants that have at least a single aromatic ring and one or more hydroxyl groups in their structures. Phenolic compounds can be divided into two classes: acids such as benzoic acid derivatives (e.g., gallic acid) and cinnamic acid derivatives such as coumaric and ferulic acid [8]. Extracted phenolic compounds from plants have potential and different applications such as antitumor, antioxidant, antimicrobial, anti-inflammatory, antiviral, skin protection from UV radiation, analgesic, and antipyretic [9,10,11,12]. Flavonoid compounds are a major group of phenolic compounds that are responsible, along with carotenoids and chlorophylls, for colors in plants, as reported by [13]. Moreover, flavonoids are highly bioactive compounds found in both edible and non-edible plants. Different solvents have been used for the extraction of phenolic compounds from plants [14,15]. For example, methanol, ethanol, acetone, water solvents, and their combinations have been used for the extraction of phenolic compounds [16,17]. One of the most effective factors in the extraction process of phenolic constituents are the polarity and type of solvent and their ratio as well as the time and temperature of extraction, chemical composition, and physical characteristics of the plant materials [18]. In accordance, extraction of phenolic compounds, flavonoids, and tannins using different solvents with different polarities such as hexane, ethyl acetate, and methanol have been used as the best solvents for polyphenols, flavonoids, and tannins, respectively [19]. Concerning the reported methods, there is a possibility of interaction between these compounds and other compounds in plants such as proteins and carbohydrates [14,15]. Hence, it would be difficult to develop an appropriate method for the extraction of all phenolic compounds [15]. Recently, substantial developments in research have focused on the extraction, detection, identification, and quantification of phenolic compounds as medicinal or biomolecules for human health. Moreover, various chemical approaches have been used to detect the existence of bioactive compounds, while spectrophotometric and chromatographic techniques have been utilized to identify and quantify individual phenolic compounds [20]. Due to the variety of phytochemical compounds contained in plant materials and their differing solubility properties in different solvents, the optimal solvent for extraction depends on the particular plant materials and the compounds that are to be isolated [21,22]. In comparing several solvents, methanol has generally been found to be more efficient in the extraction of lower molecular weight polyphenols while the higher molecular weight flavonoids are better extracted with aqueous acetone [23]. Despite the large body of literature and the investigations that have conducted, the quantification of various phenolic structural groups still remains difficult [24]. Thus, different analytical methods are urgently required for the separation and quantification of bioactive compounds such as gallic acid [25]. This study attempted to optimize the extraction method for the efficient recovery of TPC, TFC, and TTC in the leaf and seed extracts of medicinal plants (*J. procera*), separation and quantification of gallic acid, quercetin, and tannic acid besides the screening of bioactive compounds. This is the first research to have been conducted on an optimization method for the extraction, separation, quantification, and screening of bioactive compounds from the leaf and seed extract of *J. procera*. Therefore, different types of solvents with different polarities such as methanol, ethanol, acetone, and deionized water were investigated and evaluated whereas TPC, TFC, and TTC were estimated using a UV-spectrophotometer. Gallic acid, quercetin, and tannic acid in plant samples were separated and quantified using HPLC along with authentic standards. This work serves as a good basis for other researchers to estimate, separate, and quantify the phenolic compounds in medicinal plant such as *J. procera.*


## 2. Results and Discussion

### 2.1. Total Phenolic Compounds Recovery from Leaf Extract of J. procera

The optimized method for extraction phenolic compounds (TPC, TFC, and TTC) from medicinal plants such as *J. procera* has important and significant meaning to future biomolecules for human health, pharmaceutical, and medicinal research. The selection of the solvent and the conditions of the extraction process is the main step in the development of the technique for the qualitative and quantitative analysis of phytochemical compounds in plant materials. The extraction solvent is the main factor in the prognosis of the qualitative and quantitative composition of the extracted phenolic compounds. The most common solvents used for the extraction of phenolic compounds from plant materials are methanol, ethanol, acetone, and their various aqueous mixtures of various concentrations [13,26]. Hence, for an optimization method for TPC, TFC and TTC extraction from the leaf and seed extract of *J. procera*, four solvents with different polarities were used. The effect of solvents in the extraction of targeted compounds was investigated using UV-spectrophotometer and high-performance liquid chromatography (HPLC) along with reference standards (gallic acid, quercetin, and tannic acid) that were used for the calibration curves (Figure 1). First, the investigated solvents recorded significant differences in the value of extracted phenolic compounds. The value of the total phenolic content from the leaf extract ranged from 2.9 to 9.7 mg GAE/g DW (Table 1). This indicated that methanol (99.8%) is best for TPC recovery (9.7 mg) from the leaf of *J. procera*, followed by acetone (7.4 mg), deionized water (3.0 mg), and ethanol (2.9 mg), with significant differences between the investigated solvents (Table 1). The recovery yield of total flavonoid content (TFC) from the leaf extract varied from 0.9 to 5.9 mg QE/g DW (Table 1), which indicated that ethanol (99%) was the best solvent for the extraction of TFC with significant results (5.9 mg) in contrast with the other solvents, followed by methanol (3.8 mg), acetone (2.9 mg), and deionized water (0.9 mg) (Table 1). TTC from the leaf extract of *J. procera* ranged between 1.5 to 4. 3 mg TA/g DW (Table 1), which revealed that acetone (99.5%) was the best solvent for TTC extraction from the leaf of *J. procera* (3.6 mg) compared to other solvents (Table 1) categorized by acetone (3.6 mg), ethanol (1.7), and deionized water (1.5 mg). It has been reported that the methanol leaf extract of *J. procera* contained 896.5 mg/100g of TPC [27], which in agreement with our findings that TPC recovery of menthol leaf extract was 9.7/g DW. No reports were found in the literature review related to the extraction of TPC, TFC, and TTC in the leaf of *J. procera* using acetone, ethanol, and deionized water. Moreover, it has been stated that methanol is the best solvent for the extraction of phenolic compounds from plants [19,28,29]. The present results showed that different solvents in the same conditions resulted in various extraction values of phenolic compounds. It might be that the differences in the polarity of solvents could cause a wide variation in the level of extracted bioactive compounds [28].

### 2.2. Total Phenolic Compounds Recovery from Seed Extract of J. procera

Table 2 shows the efficiency and effect of different solvents in the extraction and yield recovery of phenolic compounds (TPF, TFC, and TTC) from the seed extract of *J. Procera*. The yield of TPC in the seed extract ranged from 0.53 to 2.6 mg GAE/g DW. This demonstrated that among the different solvents investigated, ethanol was the best solvent for the extraction of TPC from the seed of *J. procera* (2.6 mg), followed by acetone (1.91 mg), methanol (1.9 mg), and deionized water (0.53 mg) (Table 2). The yield of TFC in seed extract varied from 0.5 to 1.6 mg QE/g DW (Table 2), which showed that ethanol was the best solvent for yield recovery of TFC from seed extract (1.6 mg) in contrast with other solvents tracked by methanol (1.5 mg), acetone (1.3 mg), and deionized water (0.5 mg). The TTC value in seed extract of *J. Procera* was fluctuated between 0.5 and 1.4 mg TA/g DW. The results showed that methanol is the best solvent for the yield recovery of TTC (1.4 mg) followed by ethanol (1.2 mg), acetone (1.1 mg), and deionized water (0.5 mg) (Table 1). Different solvents have shown different effects in the extraction of phenolic compounds from the different parts of plants. This effect might depend on the polarity of the solvent as well as the specific part of the plant material. In the literature, no report was found related to the extraction of phenolic compounds from the seed of *J. procera.* In contrast, the effect of solvents in the extraction of phenolic compounds from leaf and seed is different according to the part of the plant as well as the polarity of the solvents. For example, organic solvent methanol was the best for TPC yield recovery from leaf, whereas ethanol achieved the highest value of total phenolic content from the seed. In accordance, it has been reported that the optimal solvent for extraction depends on the particular plant materials and the compounds that are to be isolated [21,22]. The extraction of phenolic compounds from plant materials depends mostly on the nature of the sample matrix and the chemical properties of the phenolics including the concentration, polarity, molecular structure, number of aromatic rings and hydroxyl groups reported by [20]. Moreover, different parts of a plant occupy a pool of bioactive compounds containing potential chemical groups [30]. 

### 2.3. Separation and Quantification of Gallic Acid, Quercetin, and Tannic Acid from Leaf and Seed Extract 

Gallic acid is a class of phytochemicals with powerful anti-inflammatory, anti-microbial, and anti-tumor properties [31,32,33,34]. While quercetin is a natural flavonoid with antioxidant and high biological activity [35,36], tannic acid is a specific type of plant phenolic that presents unique antibacterial as well as antiviral properties [37,38]. High-performance liquid chromatography is the recommended approach for the separation and quantification of phenolic compounds from plant materials [20]. However, different factors affect the chromatography analysis of phenolic compounds, for example, sample purification, mobile phase, column types, and detectors [39]. In the current study, the separation and quantification of gallic acid, quercetin, and tannic acid in materials of the medicinal plant (*J. procera*) were achieved using HPLC. A broad variety of analytical methods with different mobile phases (acetonitrile, methanol, acetic acid, and deionized) have been investigated for the separation of phenolic compounds (gallic acid, quercetin, and tannic acid) in plant samples. Chromatograms were acquired at four different wavelengths (254, 274, 278, and 300 nm) according to the absorption maxima of the analyzed compounds. For gallic acid resolving, 274 nm showed the best result, while for resolving tannic acid and quercetin, 278 nm was the best concerning other conditions detailed in the Methods section. Gallic acid was separated from plant materials using the chromatography method described by [40] in which the mobile phase consisted of 1% acetic acid and methanol (40:60) (*v*/*v*). This method is applicable to the separation of gallic acid in the sample with minor modifications (Figure 2). The quantification of gallic acid in plant materials was conducted using gallic acid as a reference for preparing a calibration curve. The amount of gallic acid in the leaf extract ranged from 6.6 to 9.2 µg/g DW. While in the seed extract ranged from 6.5 to 7.2 µg/g DW with significant differences among different solvents used in the extraction (Table 3 and Table 4). For quercetin separation in plant materials, we found that the mobile phase consisting of acetonitrile and methanol (40: 60) (*v*/*v*) with the conditions detailed in the Methods section was applicable for quercetin separation (Figure 3). The quercetin in plant materials was quantified using quercetin as the reference standard, the amount of quercetin in leaf extract varied from 6.3 to 18.2 µg/g DW, and in seed ranged from 0.97 to 4.2 µg/g DW, which indicated that methanol was the best solvent for the extraction of quercetin from leaf with significant differences. Ethanol showed better efficiency in the extraction of quercetin from seed. Tannic acid in plant materials was chromatographically separated using an authentic standard. Among the different methods with several types of reagents tested, methanol and 0.6% acetic acid in a combination of (20:80) (*v*/*v*) was suitable for the separation of tannic acid in plant materials (Figure 4) with conditions stated in the Methods section. The quantification of tannic acid in plant materials was undertaken using tannic acid as the reference for the calibration curve. The amount of tannic acid in the leaf extract ranged from 16.2 to 29.3 µg/g DW, and in seed extract ranged from 6.6 to 9.7 µg/g DW with significant differences among solvents (Table 3 and Table 4). It has been reported that methanol, acetic acid, and acetonitrile and their aqueous forms are the main mobile phases utilized in the HPLC separation and quantification of phenolics [39,41,42]. Although resolving some compounds was poor in the current study, the approach could serve as a basis and can be developed further. Finally, methanol was the best solvent for the extraction of gallic acid, quercetin, and tannic acid, and ethanol was the best solvent for gallic acid, quercetin, and tannic acid extraction from the seed of *J. procera*.

### 2.4. GC-MS Analysis of Seed and Leaf Extract of Juniperus procera 

Generally, plants produce secondary metabolites as a protection mechanism against biotic and abiotic stress. The identification of the phytochemical constituents in the seed and leaf extract of *J. procera* was performed using commercial libraries and a comparison of the mass spectra, match percentage, and the retention times of the reference compounds. Since the ethanol extract from seed contained a higher value of phenolic compounds, it was submitted to GC-MS analysis to identify the phytochemical compounds and resulted in very important bioactive compounds such as ferruginol, phenanthrene, and n-hexadecanoic acid related to phenolic compounds (Table 5 and Figure 5). The bioactive compounds in the leaf extract of *J. procera* were screened previously [43]; however, some variation has been recorded between the seed and leaf extract (Table 5) in the screened bioactive compounds. Moreover, different parts of the plant occupy a pool of bioactive compounds containing potential chemical groups [30]. The detected phytochemical compounds from plant materials contained antimicrobial and antitumor agents. For example, it has been reported that ferruginol is a diterpene phenol and has received attention due to its pharmacological properties including anti-tumor, antimalarial activity, antibacterial, gastro-protective, and cardio-protective effects [44,45,46]. Furthermore, the most important bioactive compounds are highlighted above. The other phytochemical constituents detected in the seed and leaf extract of *Juniperus procera* and their biological activities are presented in Table 5, Figure 5, and Appendix A. 

## 3. Material and Methods

### 3.1. Reagents and Standards

Three standards were used in this experiment: gallic acid, quercetin, and tannic acid. Deionized water and organic solvents (methanol 99.8%, ethanol 99%, and acetone 99.5% were purchased from Sigma-Aldrich. 

### 3.2. Preparation of Leaf and Seed Extract of Juniperus procera 

The leaf and seeds of *J. procera* were air dried and ground. Then, 1 g of powdered seed and the same amount from leaf were extracted using 80 mL of (methanol 99.98%, ethanol (99%), acetone (99.5%) and deionized water). The extraction process was performed in an Innova 44 Incubator Shaker at 120 rpm, at a temperature of 28 ± 2 °C for 24 h. The aqueous and organic phases were separated by centrifugation at 5000 rpm for 15 min. Then, the organic phase was collected and evaporated in a vacuum. The residues were reconstituted with 2 mL of methanol and filtered with 0.45 µm nylon syringe before estimated UV-spectrophotometer (SHIMADZU, UV − 1800, Japan) and high-performance liquid chromatography (HPLC). 

### 3.3. Estimation of the Total Phenolic Content

The total phenolic compounds in the leaf and seed extract of *J. procera* were estimated using Folin–Ciocalteu reagent following the method described by Ainsworth with some modifications [67]. A volume of 50 µL of the plant material extract was mixed with 50 µL of the Folin–Ciocalteu reagent and 1.5 mL of deionized water for 8 min. This was then neutralized with 50 µL of sodium carbonate solution (20%). The reaction mixture was incubated at room temperature for 30 min. Gallic acid was used as a reference standard (100, 150, 300, 400, 600, and 1000 µg/mL). The absorbance of the resulting blue color was measured at 765 nm using a UV-spectrophotometer (SHIMADZU, UV-1800, Japan). The total phenolic content was estimated from the linear equation of a standard curve prepared with gallic acid (Y = 0.0033 + 0.0752 with R^2^ = 0.9855) Figure 1a. The content of total phenolic compounds was expressed as mg/g gallic acid equivalent (GAE) of dry weight.

### 3.4. Estimation of the Total Flavonoid Content 

Estimation of the total flavonoid content in the leaf and seed extract of *J. procera* was carried out using the method described by [68]. A volume of 0.2 mL of 2% AlCl_3_ was added to 0.2 mL of plant material extract in a 2 mL tube. After one hour at room temperature, 0.4 mL of deionized water was added to the solution. The absorbance was measured at 420 nm. A calibration curve was obtained using the quercetin reference standard (100, 200, 400, 600, and 800 µg/mL). Total flavonoid content was expressed as quercetin (mg/g DW) using the following equation (y = 0.0042x − 0.1673 with R^2^ = 0.9871 ) based on the calibration curve (Figure 1b). 

### 3.5. Estimation of Total Tannin Content

The total tannin content in leaf and seed extract from *J. procera* was estimated using the Folin–Ciocalteu method described by [69] with minor modifications. A total of 50 µL of the plant extract was added to a tube (2 mL) containing 1.5 mL of deionized water and 50 µL of Folin–Ciocalteu phenol reagent for 8 min. Then, 50 µL of 35% sodium carbonate solution was added to the mixture. The mixture was shaken well and kept at room temperature in the dark for 20 min. Tannic acid was used as the reference, so standard solutions of tannic acid (250, 500, 80, 750 μg/mL) were prepared (Figure 1c). The absorbance of the samples and standard solutions was measured with a UV/Visible spectrophotometer (SHIMADZU, UV-1800, Japan) against the blank that consisted of 50 µL of Folin–Ciocalteu phenol reagent, 1.5 mL deionized water, and 50 µL of sodium carbonate 35% at 700 nm. The estimation of the total tannin content (TTC) was carried out in triplicate using the following equation (Y = 0.0054−0.0252 with R^2^ = 9937). The total tannin content was expressed in terms of mg/g DW.

### 3.6. HPLC Equipment

An Agilent liquid chromatographic system-USA controlled by G 4226A software with the column SB-C18 (1.8 μm, 4.6 × 150 mm) was used for the separation and quantification of gallic acid, quercetin, and tannic acid in plant materials.

### 3.7. Chromatographic Analysis of Gallic Acid

The mobile phase used for the separation and quantification of gallic acid in the sample consisted of 1% aqueous acetic acid solution (A) and methanol (B) (40:60) (*v*/*v*). Samples were eluted with the following gradient: flow rate of 0.700 mL/min, and injection volume of 1 µL. The column temperature was maintained at 25 °C. The chromatogram was acquired at a wavelength of 274 nm according to the absorption maxima of the analyzed samples (Figure 2). The gallic acid sample was identified by its retention time and by spiking with gallic acid as the standard under the same conditions [40]. The gallic acid in plant materials was estimated from the linear equation (y = 4722.3x − 668.15, R^2^ = 0.968) of a standard curve prepared with gallic acid (250, 500, 1000 µg/mL). 

### 3.8. Chromatographic Analysis of Quercetin 

The mobile phase used for the separation of tannic acid in plant materials consisted of aqueous acetic acetonitrile (A) and methanol (B) (40:60) (*v*/*v*). Flow rate was 0.700 mL/min ((518.88 bar) and the injection volume was 1 µL. The column temperature was maintained at 25 °C. The chromatogram was acquired at wavelengths of 278 nm according to the absorption maxima of the analyzed sample. The quercetin was identified by its retention time and by spiking with quercetin as the reference standard under the same conditions (Figure 3). The quercetin in plant materials was quantified from the linear equation (y = 1146.7x − 45.816, R^2^ = 0.9969) prepared from the authentic standard (250, 500, and 1000 µg/mL).

### 3.9. Chromatographic Analysis of Tannic Acid

For tannic acid separation using HPLC, the mobile phase consisted of 0.6% acetic acid solution (A) and methanol (B) (20:80) (*v*/*v*). Flow rate was 1 mL/min (361.74 bar) and the injection volume was 1 µL. The column temperature was adjusted at 28 °C. The chromatogram was acquired at a wavelength of 278 nm according to the absorption maxima of analyzed sample. The tannic acid in plant materials was identified by its retention time and by spiking with tannic acid as the standard under the same conditions of separation (Figure 4). The tannic acid in plant materials was estimated from the linear equation (y = 1011x − 694.17, R^2^ = 0.9957) of a standard curve prepared with tannic acid (1000, 1500, and 2000 µg/mL).

### 3.10. Preparation of Plant Materials for GC–MS Analysis

Based on the results of the UV-spectrophotometer and ANOVA test, the ethanol seed extract was selected to be injected into GC-MS analysis for bioactive compound profiling by using gas chromatography-mass spectrometry (GC-MS 7890A; Agilent Technologies-USA, equipped with a 5975 mass-selective detector and a 7693 automated liquid sampler, fitted with a DB-5MS GC column (30 m length, 0.25 mm inner diameter, and 0.25 μm film thickness)). The extract was filtered using a 2 µm membrane filter. Then, a 1.0 µL aliquot of the ethanol extract was injected into the system. The injection temperature was 280 °C and the column temperature was adjusted to 300 °C. Helium gas was used as the carrier with a flow rate of 1 mL/min. The electron ionization energy was 70 eV while the GC-MS analysis leaf extract of *J. procera* was undertaken and published recently [43]. 

### 3.11. Statistical Analysis

Experiments were conducted in triplicate while the outcomes reported in the tables and figures were the average of three replicate  ±  standard deviations. Using SPSS (version 20) software, one-way ANOVA was performed to evaluate the statistical significance at *p* < 0.05.

## 4. Conclusions 

In summary, this present study investigated the effect and efficiency of different solvents on the extraction of phenolic compounds in the leaf and seed of *J. procera.* Methanol was the best solvent for the extraction of TPC among the tested solvents. Ethanol achieved the highest TFC value from the leaf extract and acetone the highest TTC recovery from the leaf extract of *J. procera*. In the case of the seed extract of *J. procera,* ethanol was the best solvent for the extraction of TPC and TFC content, in contrast with other solvents. Methanol was the best solvent for the yield recovery of TTC from the seed of *J. procera*. Additionally, in this study, gallic acid, quercetin, and tannic acid in the plant materials were chromatographically separated and quantified using HPLC. Moreover, bioactive compounds in the seed and leaf extract of *J. procera* were identified using GC-MS analysis. Obviously, solvents have shown a significant effect in the extraction of phenolic compounds. Leaf extract of *J. procera* contained higher phenolic compounds than the seed extract with a significant difference. We concluded that the optimal solvent for extraction depends on the particular plant material and the compounds that are to be isolated. The specific mobile phases are very important for the separation of phytochemicals using HPLC. The approach developed and reported in this work can be applied to the identification, determination, and evaluation of bioactive compounds in medicinal plants such as *J. procera*. Furthermore, the bioactivity of detected compounds should be investigated. 

## Figures and Tables

**Figure 1 molecules-26-07454-f001:**
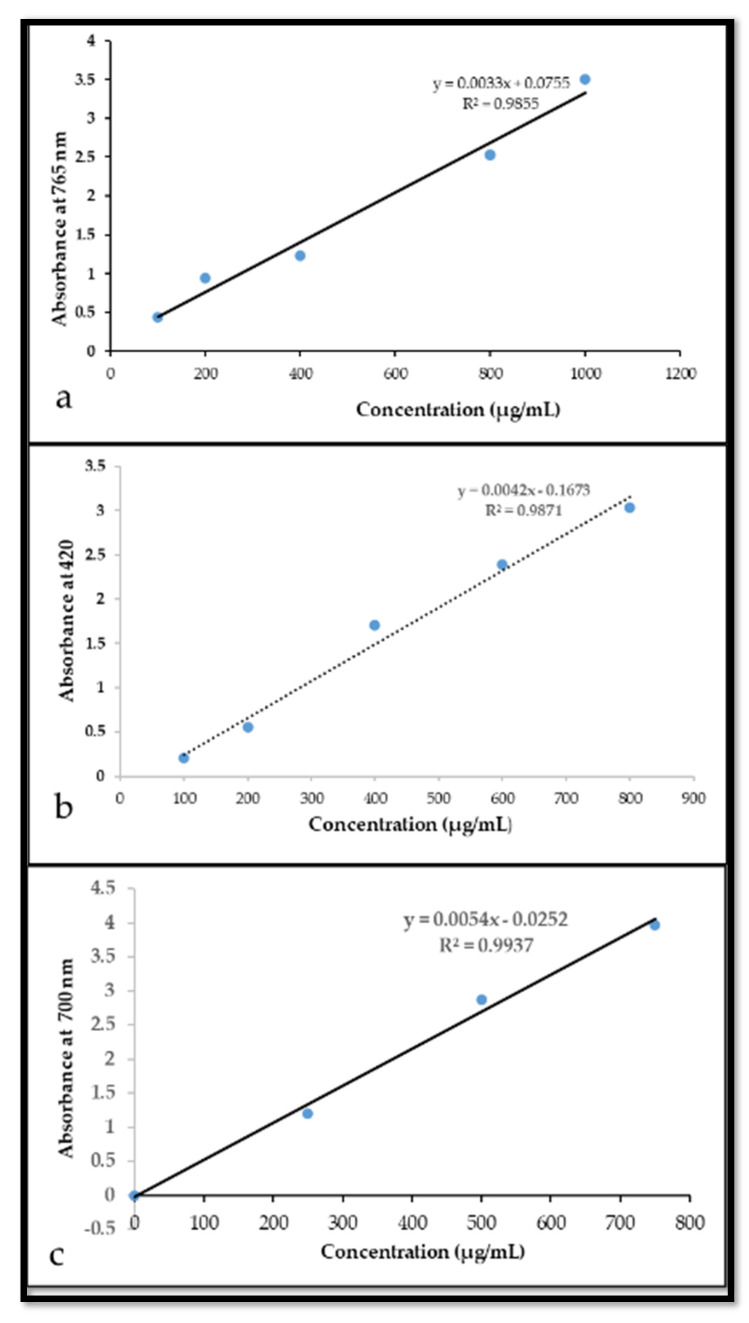
Shows calibration curves prepared from authentic standards (**a**) gallic acid, (**b**) quercetin, (**c**) tannic acid.

**Figure 2 molecules-26-07454-f002:**
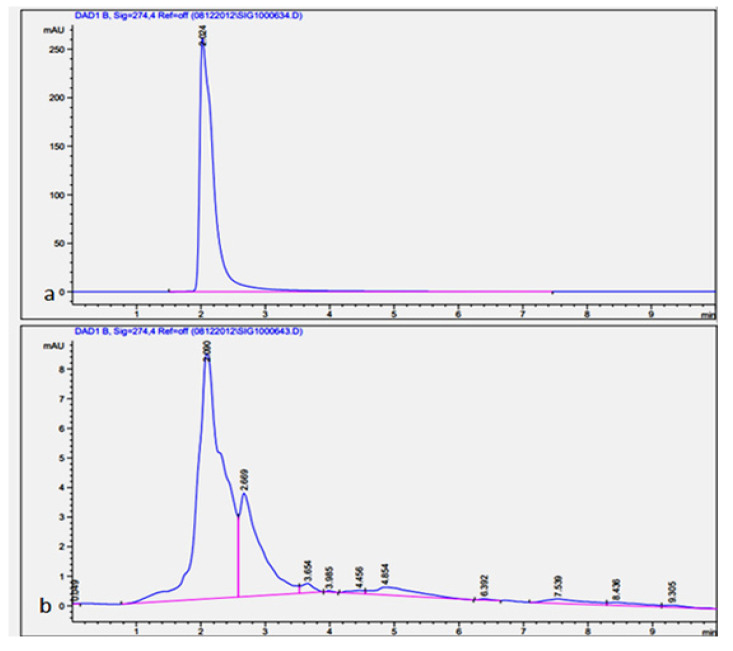
(**a**) HPLC chromatogram of the reference standard of gallic acid (at 274 nm). (**b**) HPLC chromatograms of the phenolic compound in the plant extract (at 274 nm) shows a retention time at 2.0 min.

**Figure 3 molecules-26-07454-f003:**
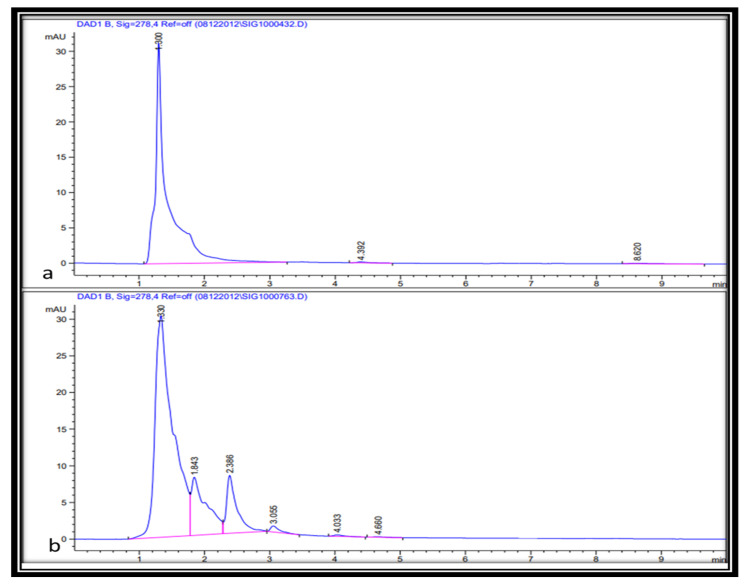
(**a**) HPLC chromatogram of the quercetin standard (at 278 nm). (**b**) HPLC chromatogram of the flavonoid compound in the plant extract (at 278 nm) shows a retention time at 1.3 min.

**Figure 4 molecules-26-07454-f004:**
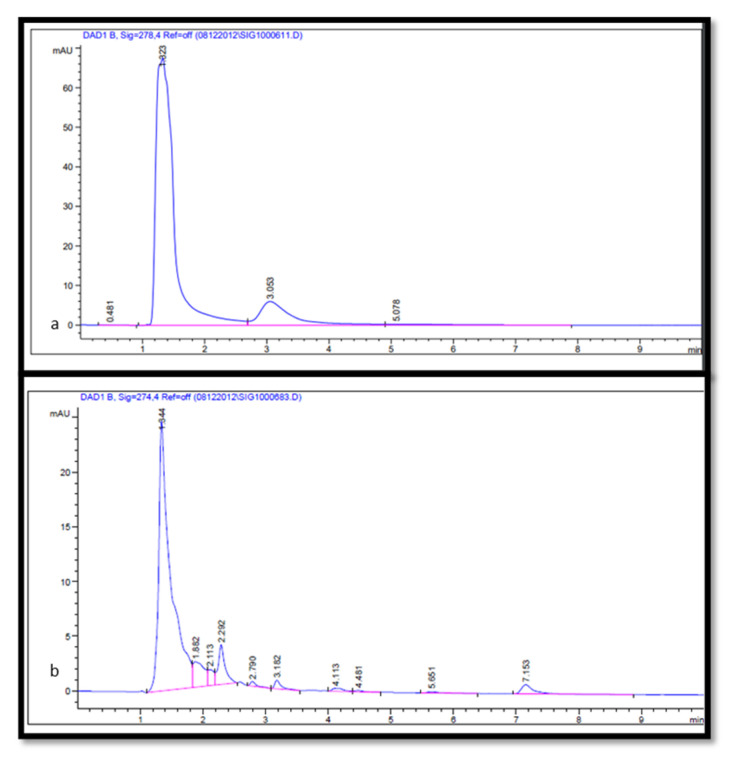
(**a**) HPLC chromatogram of the tannic acid standard (at 274 nm). (**b**) HPLC chromatogram of the tannin compound in the plant extract (at 274 nm) shows retention time at 1.323 min.

**Figure 5 molecules-26-07454-f005:**
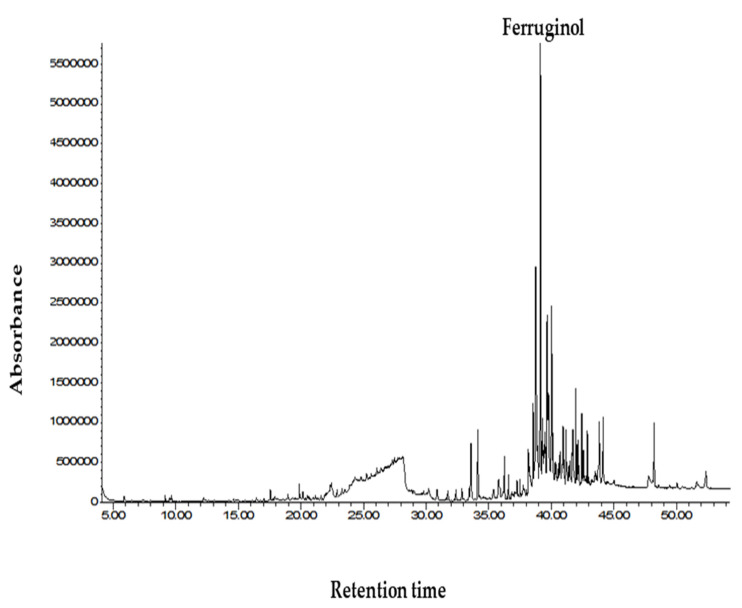
GC chromatogram of the ethanol seed extract of *Juniperus procera*.

**Table 1 molecules-26-07454-t001:** Effect of different solvents on the extraction of TPC, TFC, and TTC from leaf extract of *J. procera* (mg/g DW).

Solvents	TPC	TFC	TTC
Methanol (99.8%)	9.7 ± 0.04 ^a^	3.8 ± 0.05 ^b^	3.6 ± 0.10 ^b^
Acetone (99.5%)	7.4 ± 0.01 ^b^	2.9 ± 0.12 ^c^	4.3 ± 0.98 ^a^
Deionized water	3.0 ± 0.07 ^c^	0.9 ± 0.09 ^d^	1.5 ± 0.08 ^d^
Ethanol (99%)	2.9 ± 0.08	5.9 ± 0.03 ^a^	1.7 ± 0.6 ^c^

The data are presented as the average of total phenolic content, total flavonoid content, and total tannic content in leaf extract ± standard deviation (SD). ^a,b,c,d^ Means within the same column with different superscripts differ significantly at (*p* < 0.05).

**Table 2 molecules-26-07454-t002:** Effect of different solvents on the extraction of TPC, TFC, and TTC from the seed extract of *J. procera* (mg/g DW).

Solvents	TPC	TFC	TTC
Methanol (99.8%)	1.90 ± 0.17 ^b^	1.5 ± 0.03 ^b^	1.4 ± 0.19 ^a^
Acetone (99.5%)	1.91 ± 0.81 ^b^	1.5 ± 0.05 ^c^	1.1 ± 0.06 ^b^
Deionized water	0.53 ± 0.17 ^c^	0.5 ± 0.09 ^d^	0.5 ± 0.03 ^c^
Ethanol (99%)	2.6.13 ± 0.06 ^a^	1.6 ± 0.07 ^a^	1.2 ± 0.08 ^b^

The data are presented as the average of total phenolic content, total flavonoid content, and total tannic content in seed extract ± standard deviation (SD). ^a,b,c,d^ Means within the same column with different superscripts differ significantly (*p* < 0.05).

**Table 3 molecules-26-07454-t003:** Effect of different solvents on extraction of gallic, quercetin, and tannic acid from the leaf extract of *J. procera* (µg/g DW).

Solvents	Gallic Acid	Quercetin	Tannic Acid
Methanol (99.8%)	9.2 ± 0.13 ^a^	18.2 ± 0.25 ^a^	29.3 ^a^ ± 0.2 ^a^
Acetone (99.5%)	8.0 ^b^ ± 0.06 ^b^	16.4 ± 0.37 ^c^	16.7 ± 0.13 ^c^
Deionized water	6.6 ± 0.20 ^c^	6.3 ± 0.18 ^d^	16.2 ± 0.33 ^d^
Ethanol (99%)	8.0 ^b^ ± 0.05 ^b^	17.2 ± 0.25 ^b^	17.1 ± 0.13 ^b^

The data are presented as the average of gallic, quercetin, and tannic acid content in leaf extract ± standard deviation (SD). ^a,b,c,d^ Means within the same column with different superscripts differ significantly at (*p* < 0.05).

**Table 4 molecules-26-07454-t004:** Effect of different solvents on the extraction of gallic, quercetin, and tannic acid from the seed extract of *J. procera* (µg/g DW).

Solvents	Gallic Acid	Quercetin	Tannic Acid
Methanol (99.8%)	6.7 ± 0.26 ^b^	3.6 ± 0.25 ^b^	8.7 ± 0.06 ^b^
Acetone (99.5%)	6.5 ± 0.01 ^d^	1.8 ± 0.20 ^c^	6.6 ± 0.60 ^d^
Deionized water	6.6 ± 0.26 ^c^	0.97 ± 0.25 ^d^	6.7 ± 2.40 ^c^
Ethanol (99%)	7.2 ± 0.26 ^a^	4.2 ± 0.01 ^a^	9.3 ± 0.40 ^a^

The data are presented as the average of gallic, quercetin, and tannic acid content in the seed extract ± standard deviation (SD). ^a,b,c,d^ Means within the same column with different superscripts differ significantly at (*p* < 0.05).

**Table 5 molecules-26-07454-t005:** Major and minor compounds in the seed and leaf extract of *Juniperus procera* detected by GC-MS and their biological activity.

Seed Extract-Compounds	Retention Time	Molecular Formula	Molecular Weight (g/mol)	Bioactivity	Leaf Extract-Compound
1,3-Dioxolane	22.357	C_3_H_6_O_2_	74.08	Antifungal and antibacterial [47]	The bioactive compounds in leaf extract of *J. procera* were published recently [43,48]
Thiophene	26.073	C_4_H_4_S	84.14	Analgesic and anti-inflammatory [49]	
Heptanoic acid	22.433	C_7_H_14_O_2_	130.18	Anti-prostate cancer activity [50]	
Phosphine	32.390	H_3_P or PH_3_	33.998	Fumigant [51] and toxic [52]	
n-Hexadecanoic acid	30.889	C_21_H_46_O_2_Si_2_	386.8	Anti-inflammatory [53]	
Phenanthrene	32.860	C_14_H_10_	178.23	Anti-inflammatory, antiallergic, antimicrobial, cytotoxic, antiplatelet aggregation and phytotoxic [54,55,56,57]	
Kaur-16-ene	34.143	C_20_H_32_	272.5	Analgesic and Anti-inflammatory [58]	
Adamantane	35.410	C_10_H_16_	136.23	Antimicrobial [59]	
Phthalic acid	35.897	C_8_H_6_O_4_	166.13	Plasticizers [60]	
Ferruginol	38.908	C_20_H_30_O	286.5	Antibacterial, antimalarial and antitumoral [56,57,61]	
Palmitoyl chloride	41.500	C_16_H_31_ClO	274.9	Antioxidant activity [62]	
1,2-Benzenedicarboxylic acid	41.970	C_8_H_6_O_4_	166.14	Antimicrobial [63]	
2,6-Phenanthrenediol	42.440	n/a	n/a	Anti-inflammatory [64]	
9(1H)-Phenanthrenone	42.574	n/a	n/a	Antifungal and anti-inflammatory [65]	
1H-Indene	47.758	C_19_H_36_	264.4891	Anti-inflammatory [66]	
Beta-Sitosterol	52.355	C_29_H_50_O	414.7	Inhibits HT-29 human colon cancer	
Gamma-Sitosterol	52.355	C₂₉H₅₀O	414.386	Biomolecule for human health [62]	

## Data Availability

The data used or analyzed in this present study are available from corresponding author.

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
