# Peer review of "Optimization Method for Phenolic Compounds Extraction from Medicinal Plant (Juniperus procera) and Phytochemicals Screening"

_molecules, 2021, doi:10.3390/molecules26247454_

Round 1
Reviewer 1 Report
The article is interesting. It is very well written and very organized. However, minor changes are required
- Some important article should added for literature review
- https://www.sciencedirect.com/science/article/pii/S2215017X20302939?via%3Dihub
GC/MS analysis of Juniperus procera extract and its activity with Silver Nanoparticles against Aspergillus flavus growth and aflatoxins production
- Resolutions of peaks 2b and 3b is very poor, mobile phase changes should be done for improvements or explanation should be added in discussion
- Correlation coefficient in figure 1 b is 0.97 , more points should be added for improving this value or explanation should be added in discussion
- In line 196, acetic acid in combination of (20: 80) was
Add V/V for clarifications
- Why the authors choose 287 nm and not 254 nm for UV methods ?, answer should be added in in discussion ?
- Figure 5. GC-MS Chromatogram of ethanol seed extract of Juniperus procera.
It is GC chromatogram and no Mass spectra were displayed, correct it please
Mass spectra data should be available as supplementary file
- Methods about GC/MS should be added in 3.9 section, how the authors identify the compounds in table 5 should be illustrated, which software was used ??
Best wishes
Author Response
Dear Reviewer,
Thank you very much. We appreciate the effort that you have dedicated to providing feedback on our manuscript. A point-by-point response to your comments is attached
All the best,

Reviewer 2 Report
In this work, the authors investigated the effect and efficiency of different solvents on extraction of phenolic compounds in leaf and seed of J. procera. Methanol was the best solvent for extraction of TPC among the tested solvents. Ethanol was achieved highest TFC value from leaf extract and acetone was highest TTC recovery from leaf extract of J. procera. Gallic acid, quercetin and tannic acid in the plant materials were chromatographically separated and quantified using HPLC. Moreover, bioactive compounds in seed and leaf extract of J. procera were identified using GC-MS analysis. Obviously, Solvents have showed significant effect in extraction of phenolic compounds. Leaf extract of J. procera contained higher amount of phenolic compounds than seed extract with significant difference. The work is interesting, but there are some problems that need to be revised.
- Figures 2-4 are a bit blurry, the author should have provided higher resolution images.
- In Tables 1-4, how many replicates are there in these experiment?
Author Response
Dear Reviewer
Thank you very much for the feedback we really appreciate that. please find herewith attach a point by point response to your comments.
All the best,
